# The Chemistry, Biochemistry and Pharmacology of Marine Natural Products from *Leptolyngbya,* a Chemically Endowed Genus of Cyanobacteria

**DOI:** 10.3390/md18100508

**Published:** 2020-10-06

**Authors:** Yueying Li, C. Benjamin Naman, Kelsey L. Alexander, Huashi Guan, William H. Gerwick

**Affiliations:** 1Key Laboratory of Marine Drugs, Chinese Ministry of Education, School of Medicine and Pharmacy, Ocean University of China, Qingdao 266003, China; lyy662608@gmail.com; 2Center for Marine Biotechnology and Biomedicine, Scripps Institution of Oceanography, University of California, San Diego, CA 92093, USA; bnaman@ucsd.edu (C.B.N.); k2alexan@ucsd.edu (K.L.A.); 3Li Dak Sum Yip Yio Chin Kenneth Li Marine Biopharmaceutical Research Center, Department of Marine Pharmacy, College of Food and Pharmaceutical Sciences, Ningbo University, Ningbo 315800, China; 4Department of Chemistry and Biochemistry, University of California, San Diego, CA 92093, USA; 5Skaggs School of Pharmacy and Pharmaceutical Sciences, University of California, San Diego, CA 92093, USA

**Keywords:** *Leptolyngbya*, cyanobacteria, secondary metabolites

## Abstract

*Leptolyngbya*, a well-known genus of cyanobacteria, is found in various ecological habitats including marine, fresh water, swamps, and rice fields. Species of this genus are associated with many ecological phenomena such as nitrogen fixation, primary productivity through photosynthesis and algal blooms. As a result, there have been a number of investigations of the ecology, natural product chemistry, and biological characteristics of members of this genus. In general, the secondary metabolites of cyanobacteria are considered to be rich sources for drug discovery and development. In this review, the secondary metabolites reported in marine *Leptolyngbya* with their associated biological activities or interesting biosynthetic pathways are reviewed, and new insights and perspectives on their metabolic capacities are gained.

## 1. Introduction

Cyanobacteria, also known as ‘blue-green algae’, rank among the oldest prokaryotes found on Earth and evolved to possess remarkable capabilities of oxygenic photosynthesis and nitrogen fixation, allowing them to both adapt to and affect the planet’s environmental conditions for over two billion years [1,2,3,4]. In traditional classification schema, filamentous cyanobacteria without akinetes, heterocysts or true-branching were generally assigned to the order *Oscillatoriales* (Section III) [5]. However, one specific genus of filamentous, nonheterocystous cyanobacteria with granular surface ornamentation was ultimately named *Leptolyngbya* in 1988, and placed in the order *Pseudanabaenales* [6,7,8]. The identification of this genus was supported by phylogenetic analysis, including 16S rRNA gene sequences, which at the time showed no clear relationship with any other defined genus of cyanobacteria [9].

*Leptolyngbya* originate from a diverse range of ecological habitats, including marine, fresh water, swamps, forests, rice fields, alkaline lakes, and even the polar desert or hot desert environments [10,11,12,13]. Members of this genus possess a variety of thin (0.5–3.5 μm), long filaments, and can grow solitarily or coiled into clusters and fine mats [14,15,16]. Heterocysts and akinetes are both absent in these organisms [17]. In the taxonomic databases Algaebase [18] and CyanoDB [19] there are 138 taxonomically accepted species of *Leptolyngbya* listed.

Marine collections of *Leptolyngbya* have been reported in Japan, Panama, Gulf of Thailand, the Red Sea, Hawaii, America Samoa, and others (Table 1), which illustrates that this genus is distributed widely in the world. A breadth of scientific studies have been conducted on these samples in recent decades, including morphological and phylogenetic characterization, photosynthesis and nitrogen fixation research, examination of the associated microbial community, growth modulation and natural product chemistry research [4,9,17,20,21,22]. It is well known that cyanobacteria produce many secondary metabolites, especially when grown in different environments and conditions. Variations such as temperature, pH, dissolved phosphorous levels, nitrogen content or associated symbionts can influence the expression of different natural products [22,23]. The metabolites of cyanobacteria are known to possess potent biological activity in areas such as cytotoxicity, anti-inflammation, neuromodulatory, antibacterial, and brine shrimp toxicity [24,25]. It is possible that these observed biological activities match or mimic some of the natural ecological roles of these secondary metabolites. A limited number of *Leptolyngbya* field collections have been propagated in laboratory cultures [26,27,28]. However, they grow slowly in this environment, making for additional challenges in secondary metabolite discovery from this resource [29]. This review summarizes recent research on the secondary metabolites found in the genus *Leptolyngbya,* and covers chemical, pharmacological, and biosynthetic aspects.

## 2. Chemical Diversity of the Secondary Metabolites Isolated from *Leptolyngbya*

### 2.1. Polypeptides

In 2008, McPhail and co-workers reported the discovery of coibamide A (**1**), a cyclic depsipeptide with a high degree of *N*- and *O*-methylation from a *Leptolyngbya* species collected in Panama (Figure 1) [32]. The absolute configuration was determined by a series of chiral HPLC analyses of the amino acids resulting from the hydrolyzed peptide as well as some computational modeling. Interest in coibamide A (**1**) mounted due to its exquisitely potent low-nanomolar in vitro inhibition activity against multiple cancer cell lines, including human NCI-H460, MDA-MB-231, H292, PC-3, SF-295, mouse neuro2a, LOX IMVI, HL-60(TB), and SNB-75 cell lines [32]. The number of cell lines was expanded to include human U87-MG, SF-295 glioblastoma cells and mouse embryonic fibroblasts (MEFs), and it was further revealed that the cyclized structure was crucial for potent biological activity [52]. Several studies subsequently reported the total synthesis of coibamide A (**1**). Yao and Lim et al. completed the total synthesis of the structure originally reported for coibamide A (**2**), as well as a synthetic *O*-desmethyl analogue. However, both the ^1^H and ^13^C NMR data for the synthetic product **2** differed from those of the natural product, which indicated that the absolute configuration of this compound required revision [33,34]. Additionally, Oishi and Fujii synthesized the d-*N*-Me-Ala epimer of coibamide A (**4**) due to an epimerization of this residue during the macrocyclization process (Figure 1) [35]. In 2015, Fang and Su were able to assign the correct configuration of coibamide A (**1**) with the revision of the l-HVA and l-*N*-Me-Ala residues to the d-HVA and d-*N*-Me-Ala after total synthesis of this alternative along with its diastereomeric analogues (Figure 1) [30]. The absolute configuration of coibamide A (**1**) proposed by Fang and Su was further confirmed by McPhail and Cheong using computational methods to calculate NMR data for the conformational space occupied by several possible diastereomers and comparison with experimental values [53]. This considerable effort to resolve the structure of coibamide A (**1**) was largely motivated by the potent cytotoxicity of this molecule, and as a byproduct, provided a number of new analogues for mechanistic and pharmacological studies. More recently, Su and Fang went on to synthesize 18 new analogues (**2**, **4**, **5**–**20**) including the originally proposed structure of coibamide A (**2**) as well as the revised correct structure (**1**) to perform a structure–activity relationship study (Figure 2) [31]. However, none of the coibamide A analogues were more potent cytotoxins than the natural product, indicating the strong correlation between the observed activity, the core molecular structure and optimization of this structure through natural evolutionary processes. The only analogue that exhibited similar inhibition as natural coibamide A was the [MeAla3-MeAla6]-coibamide (**8**), which significantly suppressed tumor growth in vivo [31,33,35].

The dolastatins are an expansive and well-known series of peptidic compounds. These were named after first being discovered from the sea hare *Dolabella auricularia*, but it was later found that the mollusk accumulates these natural products from cyanobacteria in their diets [37,39,54,55]. Dolastatin 12 (**21**) is one such compound, and has interesting 4-amino-2,2-dimethyl-3-oxopentanoic acid (Ibu) and (2*S*,3*R*)-3-amino-2-methylpentanoic acid (MAP) residues in its cyclic structure (Figure 3). The macrocycle also includes an ester linkage across a 2-hydroxy-3-methylpentanoate (Hmp) residue, and is thus a depsipeptide. While this compound was originally isolated from *D. auricularia*, it was noted to resemble the *Lyngbya majuscula* metabolite majusculamide C. Dolastatin 12 (**21**) was later re-discovered from a mixed cyanobacterial assemblage of *L. majuscula* and *Schizothrix calcicola* from Guam [36,39,56,57] as well as from a *Leptolyngbya* sp. RS03 collected in the Red Sea [36]. Dolastatin 12 (**21**) may be present as a mixture of diastereomers arising from epimerization of the acid-sensitive Ibu unit during the isolation process [57]. The configuration of this Ibu residue was determined by CD and ^1^H NMR analysis after hydrolysis and purification [39]. The correct configuration was determined to be *R* by comparing the free Ibu unit to Adhpa synthetic standards [57]. In the early studies, dolastatin 12 (**21**) was shown to inhibit actin polymerization [37,58]. Dolastatin 12 (**21**) has in vitro MICs <0.05 µg/mL against human nasopharyngeal carcinoma cells and 0.08 µg/mL against human colon adenocarcinoma cells [37]. Further cytotoxicity characterization showed that dolastatin 12 has an in vitro IC_50_ > 1 µM against HeLa cells [36,56] and a ED_50_ of 7.5 × 10^−2^ μg/mL against murine P388 lymphocytic leukemia [38,39].

Ibu-epidemethoxylyngbyastatin 3 (**22**) has a similar structural backbone to dolastatin 12 (**21**) and possesses the same Ibu residue (Figure 4). However, rather than containing the MAP as in dolastatin 12, Ibu-epidemethoxylyngbyastatin 3 has a molecular mass comparatively increased by 14 Da, which results from the presence of a 3-amino-2-methylhexanoic acid (Amha) moiety instead of MAP. In vitro cytotoxicity testing showed that Ibu-epidemethoxylyngbyastatin 3 (**22**) is at least 10-fold less toxic than dolastatin 12 (IC_50_ > 1 µM) against HeLa cells [36,56].

Grassypeptolides are a series of cyclic depsipeptides first isolated from the marine cyanobacterium *Lyngbya confervoides*. These peptides contain d-amino acids, thiazoline rings, and a β-amino acid (Figure 5) [59,60]. Further research led to the isolation and description of grassypeptolides D (**23**) and E (**24**) from a Red Sea *Leptolyngbya* sp. RS03, and grassypeptolides F and G from *Lyngbya majuscula* [61,62]. Grassypeptolides D (**23**) and E (**24**) showed cytotoxic activity to both HeLa cells (IC_50_ 335 and 192 nM, respectively) and neuro2a cells (IC_50_ 599 and 407 nM, respectively) [36,56,63]. By comparing with other grassypeptolide structures and their biological activities, an initial structure–activity relationship was deduced that indicated the *N*-Me-Phe-thn-ca-Aba-thn-ca tetrapeptide motif could be the key pharmacophore of the grassypeptolides [36,56].

Loggerpeptins A–C (**25–27**) are cyclic depsipeptides with 3-amino-6-hydroxy-2-piperidone (Ahp) residues (Figure 6) that were isolated from a Florida cyanobacterial collection identified morphologically as *Leptolyngbya* sp. [40]. These compounds were screened for serine protease inhibitory activities to assess their antimetastatic effect against breast cancer cells. Loggerpeptin A (**25**) and B (**26**) were more potent than loggerpeptin C (**27**) with IC_50_s of 0.24 and 0.22 µM against bovine pancreatic chymotrypsin and 0.24 and 0.28 µM against porcine pancreatic elastase [40]. Loggerpeptin A (**25**) was 2- and 3-fold more potent than loggerpeptin B (**26**) and C (**27**) against human neutrophil elastase (HNE) [40]. All three compounds exhibited antiproteolytic activities, with IC_50_ values under 1 µM. As the major component in the collection, loggerpeptin C (**27**) was subject to a detailed molecular study [40]. Molecular docking showed that the Leu and N-terminal Thr-1, Abu and Ala residues of loggerpeptin C (**27**) were binding to the subsites S1-S4 of HNE and porcine pancreatic elastase [40].

Molassamide (**28**) is another cyclic depsipeptide (Figure 7) that was isolated along with loggerpeptins A–C (**25–27**) [40]. Compared to loggerpeptins (**25–27**), molassamide (**28**) exhibited much more potent inhibition activity against porcine pancreatic elastase, with an IC_50_ value of 50 nM, indicating the Abu residue between Ahp and Thr-1 is important to the antiproteolytic selectivity [40]. Molassamide presents similar binding patterns in molecular docking as loggerpeptin C (**27**), with the Abu and N-terminal Thr-1, Thr-2 and Ala binding in subsites S1–S4 of HNE and porcine pancreatic elastase [40]. Molassamide also inhibited elastase from cleaving the substrate CD40 in both biochemical and cellular assays, and it also inhibited ICAM-1 cleavage and downregulated elastase-induced *ICAM-1* gene expression. Overall, this profile of activity is indicative of molassamide being a promising candidate for potential treatment of breast cancer [40].

### 2.2. Simple Esters

Lumyong et al. isolated an antibacterial compound, 2-hydroxyethyl-11-hydroxyhexadec-9-enoate (**29**) (Figure 8), from *Leptolyngbya* sp. LT19 [41], as a result of screening for antibacterial activities. They showed compound **25** to be active against *Vibrio harveyi* and *V. parahaemolyticus*, with MIC values of 250–1000 and 350–1000 µg/mL. This antibacterial activity could potentially be useful to the shrimp aquaculture industry that is often burdened by the highly damaging *Vibrio* spp. pathogens [41]. However, the absolute configuration of the single stereocenter in compound **25** has not yet been determined.

Choi et al. isolated honaucins A–C (**30**–**32**) from a Hawaiian collection of *Leptolyngbya crossbyana* which possess potent anti-inflammatory and quorum-sensing (QS) inhibitory activity (Figure 9) [42]. Chemical synthesis of honaucin and a number of analogs (**30**–**32**) revealed that the each of the functional groups is critical for both of these biological activities. Further, synthetic honaucin analogues 4-bromo-honaucin A (**33**) and 4′-iodohonaucin A (**35**) were discovered to have slightly more potent activity in ﻿cellular TRAP activity than honaucin A itself (**30**), with IC_50_ values of 0.54 and 0.61 μg/mL compared to 0.63 μg/mL for **30** (Figure 9), [43]. Mechanistic pharmacological investigations of honaucin A (**30**) indicated that the molecular target(s) involves the Nrf2-ARE (Antioxidant Response Element) pathway, and specifically involves interaction with Cys residues on Keap1 when it is complexed with Nrf2. This allows Nrf2 to be transported to the nucleus, where it activates cytoprotective genes and generates an anti-inflammatory response [64]. Additionally, further investigation of bromo-honaucin A (**33**) revealed that it provides a protective effect against bone loss in RANKL-treated murine monocyte/macrophage RAW264.7 cells [43].

### 2.3. Macrolides

Leptolyngbyolides A–D (**36**–**39**), a series of 22-membered macrolides, were isolated from *Leptolyngbya* sp. collected in Okinawa, Japan. The absolute configuration of each was assigned following an asymmetric total synthesis of leptolyngbyolide C (**38**) (Figure 10) [44]. Leptolyngbyolides A–D (**36**–**39**) were screened for cytotoxic activity against HeLa S3 cells in vitro and were found to be moderately active, with IC_50_ values of 0.99, 0.16, 0.64 and 0.15 µM, respectively [44]. Furthermore, these metabolites possessed actin-depolymerizing activity with an EC_50_ of 12.6, 11.6, 26.9 and 21.5 µM, and this may represent the mechanism for the observed cellular apoptosis caused by these compounds [44].

A new macrolide, palmyrolide A (**40**), was isolated from an environmental assemblage of a *Leptolyngbya cf.* and *Oscillatoria* spp. collected from Palmyra Atoll in the Central Pacific Ocean (Figure 11). Palmyrolide A (**40**) comprises a rare *N*-methyl enamide functionality, as well as an intriguing *t*-butyl branch that likely results biosynthetically from the incorporation of malonate and three methyl groups from *S*-adenosyl-l-methionine (SAM), as shown for apratoxin [45,65]. The absolute configuration of this molecule was assigned by total synthesis of both the natural (−)-palmyrolide A (**40**) and its enantiomer, (+)-ent-palmyrolide A. This was necessitated because the *t*-butyl substituent apparently provides the lactone ester bond with resistance to hydrolysis, precluding chemical degradation studies [66,67,68]. It was speculated that cyanobacterial secondary metabolites possessing this motif, a *t*-butyl adjacent to an ester, might be stable under a wide variety of environmental conditions, similar to what was observed in the laboratory environment. Palmyrolide A (**40**) exhibited potent inhibition of calcium oscillations in murine cerebrocortical neurons and sodium channel-blocking activity in neuroblastoma (neuro2a) cells [45].

Phormidolide (**41**) is a 16-membered macrocyclic lactone polyketide-derived metabolite that was discovered from an Indonesian *Leptolyngbya* sp. (ISB3NOV94-8A) (Figure 12). This molecule has several unique structural features including a large number of hydroxy and methyl groups on the carbon backbone, several points of both *cis* and *trans* unsaturation, and a vinyl bromide at one terminus. Phormidolide (**41**) showed in vitro brine shrimp toxicity, with an LC_50_ of 1.5 µM. Multiple NMR experiments, including GHMBC, 2D INADEQUATE, and ACCORD-ADEQUATE, G-BIRDR, X-HSQMBC NMR experiments, enabled a *J*-based configuration analysis and deduction of relative configuration of stereocenters; a variable temperature Mosher ester analysis was used to assign the absolute configuration [46]. An investigation of the biosynthetic pathway for phormidolide (**41**) used a genome sequencing approach, and identified the phormidolide biosynthetic gene cluster (*phm*). The phormidolide (**41**) gene cluster was found to be of the trans-AT PKS type, which has been relatively rarely reported in cyanobacteria. This was based on finding two discrete trans-AT open reading frames along with KS-AT adaptor regions (ATd) within the PKS megasynthase. The megasynthase possesses ketosynthases, ketoreductases, KS-AT adaptor regions, dehydratases, methyltransferases, *O*-methyltransferase, enoyl-CoA hydratases, an FkbH-like domain, a pyran synthase, an NRPS-like condensation domain and an acyl carrier protein that were consistent with structure of phormidolide (**41**). The biosynthetic pathway also provided further supporting evidence for the absolute configuration of phormidolide (**41**), due to the stereospecificity of the ketoreductases observed in phm compared with known homologues [26]. However, subsequent chemical synthesis of key fragments of phormidolide revealed the need for a revision in configuration in at least one stereocenter [69]. Simultaneously, a reanalysis of the biosynthetic gene cluster suggested additional revisions in configuration were possibly required [70]. These findings stimulated a concerted computational and NMR-based re-investigation of phormidolide’s complex 3-dimensional structure, leading to a revision in several stereocenters (**42**) [71].

### 2.4. Pyrones

Kalkipyrone A (**43**) was first reported in a mixed assemblage of *Lyngbya majuscula* and *Tolypothrix* sp. (Figure 13), and was found to have potent brine shrimp toxicity (LD_50_ = 1 µg/mL) and ichthyotoxicity against *Carassius auratus* goldfish (LD_50_ = 2 µg/mL) [48]. Kalkipyrone A (**43**) and its analogue kalkipyrone B (**44**) were later found as metabolites of a *Leptolyngbya* sp. (﻿ASG15JUL14-6) collected from America Samoa (Figure 12) [47]. Kalkipyrone A (**43**) and B (**44**) were moderately toxic to a *Saccharomyces cerevisiae* strain lacking 16 ATP-binding cassette transporter pump genes (ABC16-monster strain; IC_50_ = 14.6 and 13.4 µM, respectively), while the in vitro cytotoxicity of these two natural products against H460 human lung cancer cells was somewhat more potent (EC_50_ = 0.9 and 9.0 µM, respectively) [47].

Three related γ-pyrone-containing polyketides described as yoshinones A, B1, and B2 (**45**–**47**), were isolated from *Leptolyngbya* sp. collected from Ishigaki island Okinawa, Japan (Figure 14). The absolute configuration of each compound was attempted by a modified Mosher’s method; however, these failed possibly as a result of the low amounts of samples available [49]. Thus, to assign the absolute configuration and provide sufficient amounts for further research, the absolute stereochemistry was achieved through total synthesis and comparison of NMR and chiroptical properties between the natural product and synthetic standards [72]. Yoshinone A (**45**) was found to inhibit adipogenic differentiation against 3T3-L1 cells, with an EC_50_ of 420 nM and with little cytotoxicity (IC_50_ = 63.8 µM to *S. cerevisiae* ABC16-monster cell) [47,49]. The adipogenic differentiation against 3T3-L1 cells of yoshinone B1 (**46**) and B2 (**47**) was considerably less potent, with less than 50% activity observed at tested concentrations up to 5 µM [48]. Further examination of structure–activity relationships in this drug class indicated that the position of the pyrone ring and side chain olefin are important for the inhibition of adipogenic differentiation. Further in vitro and in vivo experiments showed that yoshinone A (**45**) stimulates lactate accumulation deriving from the glycolytic system, and increases fat utilization to compensate for an insufficient energy supply [73]. These properties could possibly support the utility of yoshinone A in anti-obesity indications.

### 2.5. Polyaromatics

A series of brominated polyphenolics, crossbyanols A–D (**48**–**51**), were isolated from an extensive benthic *Leptolyngbya crossbyana* bloom in Hawaii (Figure 15) [50]. In addition to the high level of bromination, crossbyanol B–D (**49**–**51**) also have sulfated phenolic functionalities. These metabolites were suggested to play a role in the observed coral toxicity caused by the overgrowing cyanobacteria. Crossbyanol A (**48**) was found to activate sodium influx in mouse neuroblastoma (neuro2a) cells, with an EC_50_ 20 µg/mL, whereas crossbyanol B (**49**) possessed antibiotic activity, with an MIC of 2.0-3.9 µg/mL against methicillin-resistant *Staphylococcus aureus* (MRSA). The latter metabolite also showed moderately potent brine shrimp toxicity (IC_50_ 2.8 µg/mL). Crossbyanols C (**50**) and D (**51**) were not observed to have biological activity in these tests, and all four compounds were inactive as cytotoxins to H460 human lung cancer cells [50].

### 2.6. Oxazolines

A series of polar oxazolines, named leptazolines A–D (**52–55**), were isolated from the culture media of a *Leptolyngbya* sp. (Figure 16) [51]. Their planar structures were characterized by MS and NMR along with formation of acetate derivatives. Relative configuration was determined by comparison of carbon shifts with those calculated by density functional theory (DFT). Interestingly, the calculations were found to vary as a function of the computer operating system (Ubuntu 16, Windows 10, MAC Mavericks, MAC Mojave). Biological assay showed that leptazoline B (**53**) modestly inhibited the growth of PANC-1 cells, with a GI_50_ of 10 µM [51], whereas leptazoline A (**52**) with its aromatic chlorine atom did not show any significant activity to this cell line.

### 2.7. Other

#### 2.7.1. Toxins

Cyanobacteria are known to produce a variety of toxins, including those that are hepatotoxic, neurotoxic, or cardiotoxic, and which generally increase economic burdens and impact public health [23]. Cyanobacterial populations are known to sporadically grow excessively to form blooms, and in some cases these are harmful. A total of 34 species from 15 genera and five families were screened for known toxins including the neurotoxic saxitoxin (**56**) and the hepatotoxic microcystins (e.g., microcystin-LR, **57**) (Figure 17) [17,23,74]. *Leptolyngbya* collections from the Red Sea had on average 58.9 μg/g dry wt. and 438–489 μg/g dry wt. of these two toxin classes, respectively. Toxin production by marine *Leptolyngbya* poses toxicological risks to marine organisms that may feed on them, or that may be exposed to the cyanotoxins present in seawater [74].

#### 2.7.2. Non-Toxic Metabolites

Non-toxic secondary metabolites from cyanobacteria include various chemical classes such as phytohormones, siderophores, and UV-absorbing compounds such as mycosporine amino acids (MAAs) and scytonemin (**58**); all of these have been reported in *Leptolyngbya* sp. (Figure 18) [23,75,76]. These latter two series of compounds have been shown to protect photosynthetic cyanobacteria from solar UV damage [23]. An investigation of UV-B photoprotective compounds in marine *Leptolyngbya* discovered shinorine (**59**) (Figure 18), which is now realized to be one of the most dominant MAAs present in several species of cyanobacteria [75]. Scytonemin has a broader absorption profile than the MAAs, protecting against the solar irradiance damage across the UV (UV-A, -B and -C; 250–425 nm). Interestingly, scytonemin also shows interesting anti-inflammatory activity through inhibition of polo-like kinase stimulated cell proliferation pathways [77]. The biosynthesis of scytonemin, deriving from the assembly of tyrosine and tryptophan derived components, has been studied at the genomic and mechanistic level in several studies; however, all of these studies have been conducted in other genera of cyanobacteria such as *Nostoc punctiforme* ATCC 29133 and *Lyngbya aestuarii* [78,79,80,81]. 

#### 2.7.3. Phenolic Compounds

Phenolic compounds, including flavonoids and lignans, are typically natural antioxidants as well as an important group of bioactive compounds [82]. The extract from a thermophilic cyanobacterium *Leptolyngbya* sp. collected from northern Tunisia was screened by HPLC and showed the presence of 25 phenolic compounds—gallic acid (**60**), hydroxytyrosol (**61**), protocatechuic acid (**62**), vanillic acid (**63**), isovanillic acid (**64**), 3-hydroxybenzoic acid (3-HBA) (**65**), 4-hydroxybenzoic acid (4-HBA) (**66**), resorcinol (**67**), naphtoresorcinol (**68**), syringic acid (**69**), catechol (**70**), and oleuropein (**71**) (Figure 19); chlorogenic acid (**72**), dihyrdro-*p*-coumaric acid (**73**), dihyrdro-*m*-coumaric acid (**74**), ferulic acid (**75**), and rosamerinic acids (**76**) (Figure 20); catechin (**77**), luteolin-7-glucoside (**78**), apigenin-7-glucoside (**79**), flavone (**80**), naringenin (**81**), luteolin (**82**), and apigenin (**83**) (Figure 21); resveratrol (**84**) and pinoresinol (**85**) (Figure 22)—demonstrating that *Leptolyngbya* may constitute a rich source of antioxidant natural products [83,84]. These compounds also have inherent UV-absorbing properties, albeit at more restricted wavelengths than scytonemin (**58**).

#### 2.7.4. Odorous Metabolites

Geosmin (**86**) and 2-methylisoborneol (**87**) are two earthy-musty odorous terpenoid secondary metabolites that were first isolated from actinomycetes (Figure 23). Later, these same molecules were found to have a significant presence in many cyanobacteria species. These simple terpenoids have each been reported numerous times as being among the main causes for off-flavors in water and other products [85]. Wang et al. isolated **86** and **87** from *Leptolyngbya bijugata* strains, and quantified each at 13.6–22.4 and 12.3–57.5 μg/L, respectively, demonstrating their production by *Leptolyngbya* [86].

#### 2.7.5. Pigments

Phycocyanin is a pigment–protein complex found in cyanobacteria and eukaryotic algae that functions as a light-harvesting pigment. It is widely used in biotechnological, food and pharmaceutical industries [87]. Schipper et al. analyzed the phycocyanin content in *Leptolyngbya* sp. ﻿QUCCCM 56 from a desert environment to reveal that it possesses higher and purer phycocyanin compared to the current commercial source, *Arthrospira platensis* [87]. Other than absorbing light, phycocyanin also shows potential anti-aging and proteostasis-suppressive activities. With phycocyanin treatment, the life span of wild-type (N2) *C. elegans* was extended from 14.8 to 19.1 days [88].

## 3. Laboratory Cultivation

Despite many attempts, only a few environmental collections of *Leptolyngbya* have been propagated under laboratory culture conditions and, from these, there has been a relatively low rate of natural product isolation. Rather, most of the compounds reported in this genus have been discovered from environmental samples (Table 1). Martins et al. screened five *Leptolyngbya* strains among 28 cyanobacteria samples collected from the Portuguese Coast and cultured in Z8 medium, and showed cytotoxic activities of the extracts against multiple human tumor cell lines [15,16,89]. The most commonly used media for cyanobacteria culture is BG11 [53], originally formulated in 1988 to possess synthetic sea salt, microelements mixture, deionized water and vitamin mixtures [28]. In most cases, even with suitable media and abiotic factors, filamentous cyanobacteria are slow-growing life forms, normally with growth rates much slower than algae and other bacteria, thereby presenting a challenge for secondary metabolite discovery due to the small quantities of biomass produced in cultures [28,90]. The 2-hydroxyethyl-11-hydroxyhexadec-9-enoate, by contrast, was able to be isolated from the laboratory culture of *Leptolyngbya* with the relatively small biomass of 383.6 g because of its reduced structural complexity and higher production yield [41]. 

## 4. Conclusions

*Leptolyngbya* is a widely distributed genus of cyanobacteria that has emerged to be a very rich source of structurally novel and biologically active natural products. However, to date, this genus appears to be underexplored for its chemical, biological and biosynthetic potential when compared to some other genera of filamentous cyanobacteria, such as *Moorena* and *Symploca* [90]. Working with *Leptolyngbya* has been challenging due to the difficulty in bringing it into culture in the laboratory environment. This has impeded not only new compound discovery, but also exploration of its genomic characteristics, including those that are responsible for natural product biosynthesis. It is possible that developments with the heterologous expression of cyanobacterial natural product pathways will enable a more extensive exploration of the rich secondary metabolome of this genus in the future [91,92].

## Figures and Tables

**Figure 1 marinedrugs-18-00508-f001:**
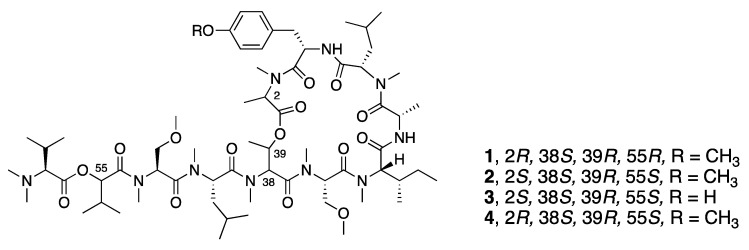
Structure of coibamide A (**2**) from *Leptolyngbya* sp. and some synthetic analogues generated to validate its absolute configuration.

**Figure 2 marinedrugs-18-00508-f002:**
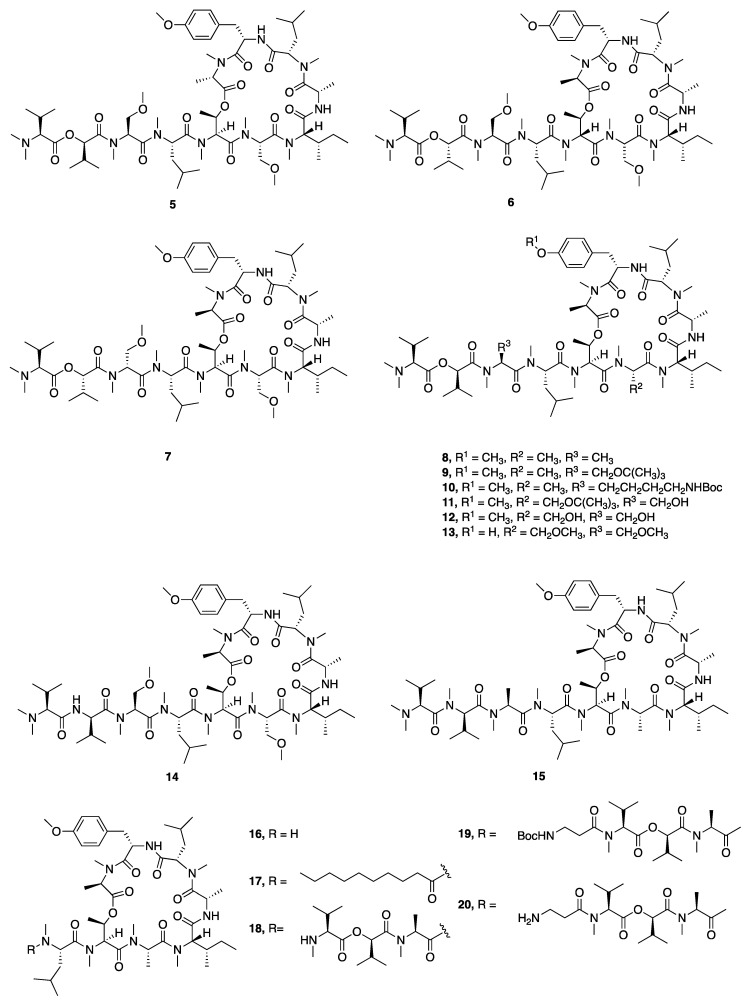
Synthetic analogues of coibamide A produced to explore structure–activity relationships (SAR) in this molecular class.

**Figure 3 marinedrugs-18-00508-f003:**
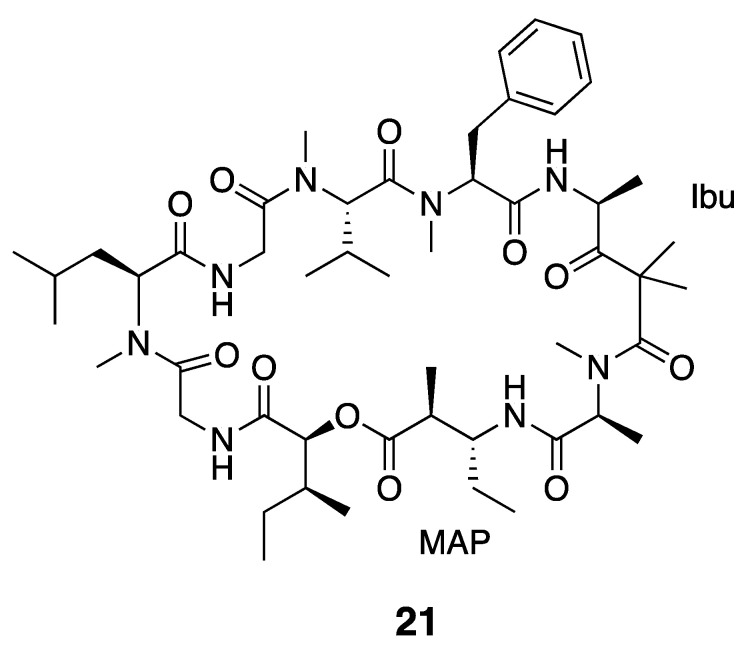
Dolastatin 12, previously isolated from *Leptolyngbya* sp. as well as several other sources. See text for discussion of the unusual MAP and Ibu moieties in dolastatin 12.

**Figure 4 marinedrugs-18-00508-f004:**
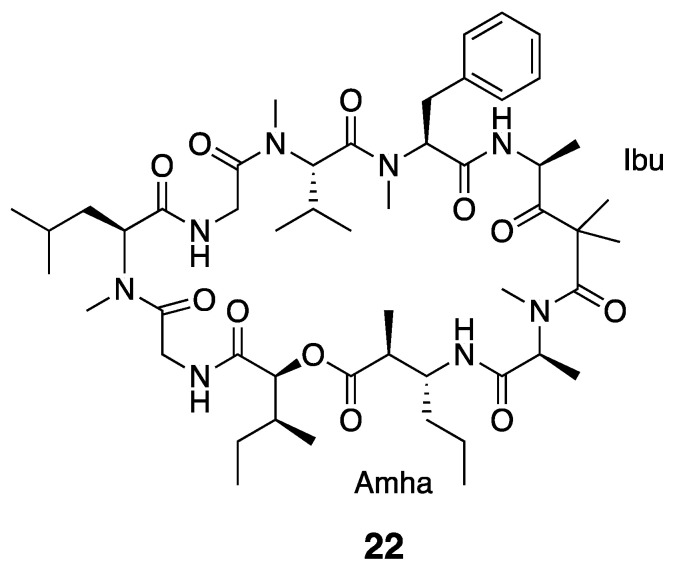
Ibu-epidemethoxylyngbyastatin 3 from *Leptolyngbya* sp.

**Figure 5 marinedrugs-18-00508-f005:**
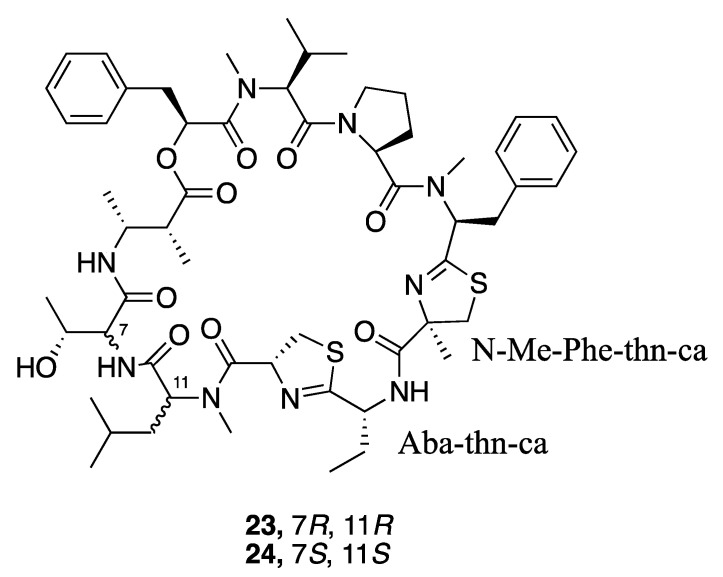
Grassypeptolides from *Leptolyngbya* sp. RS03.

**Figure 6 marinedrugs-18-00508-f006:**
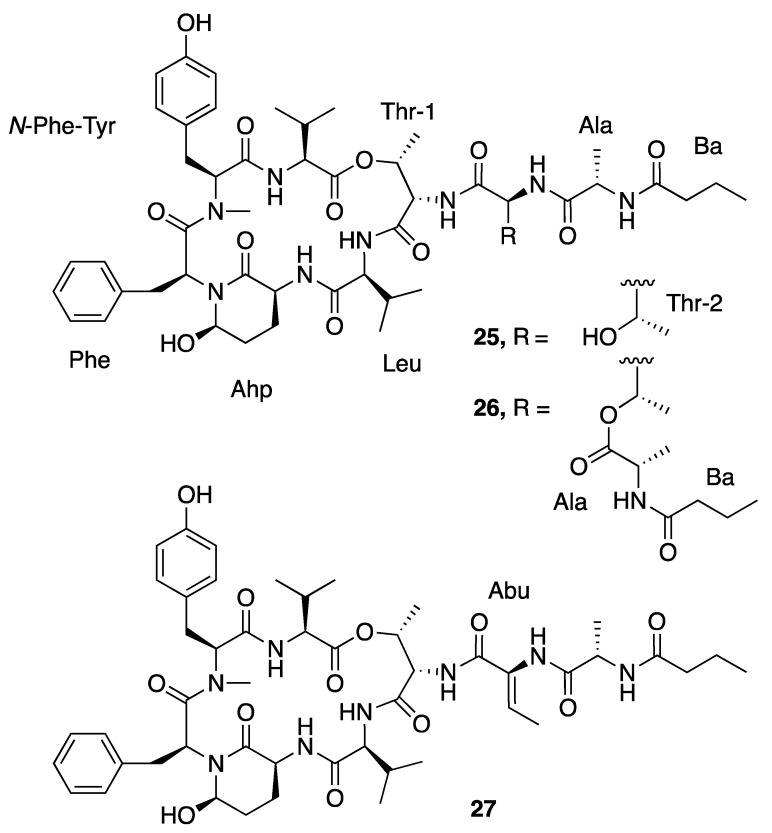
Loggerpeptins from *Leptolyngbya* sp.

**Figure 7 marinedrugs-18-00508-f007:**
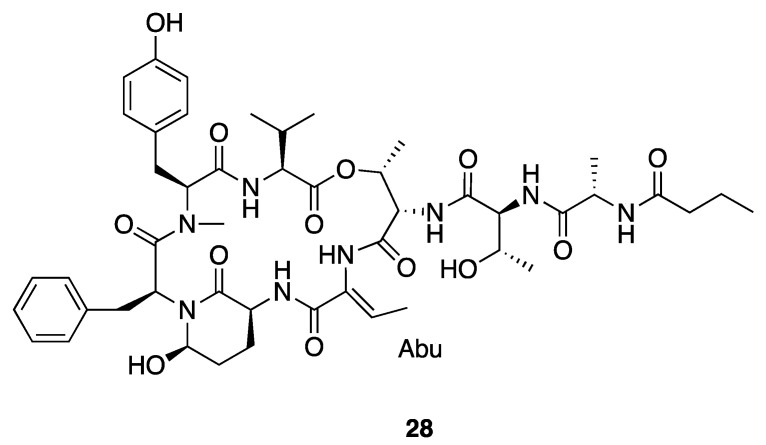
Molassamide from *Leptolyngbya* sp.

**Figure 8 marinedrugs-18-00508-f008:**
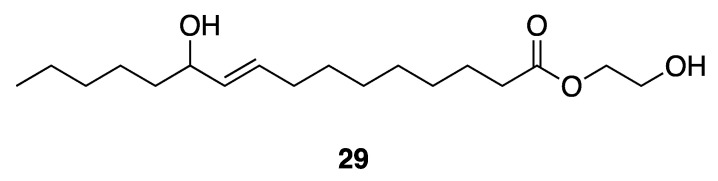
2-Hydroxyethyl-11-hydroxyhexadec-9-enoate from *Leptolyngbya* sp. LT19.

**Figure 9 marinedrugs-18-00508-f009:**
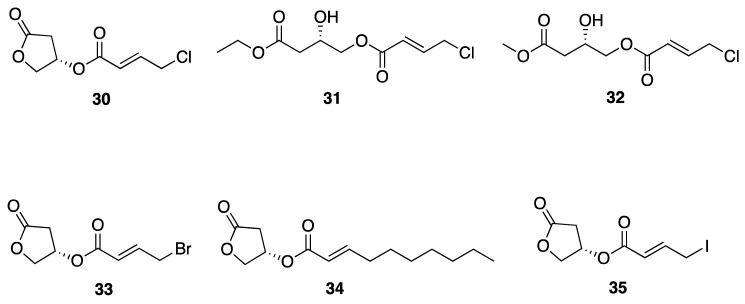
Honaucins A–C from *Leptolyngbya crossbyana* and some synthetic derivatives.

**Figure 10 marinedrugs-18-00508-f010:**
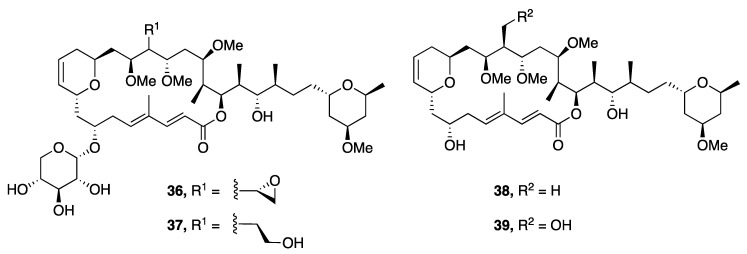
Leptolyngbyolides from *Leptolyngbya* sp.

**Figure 11 marinedrugs-18-00508-f011:**
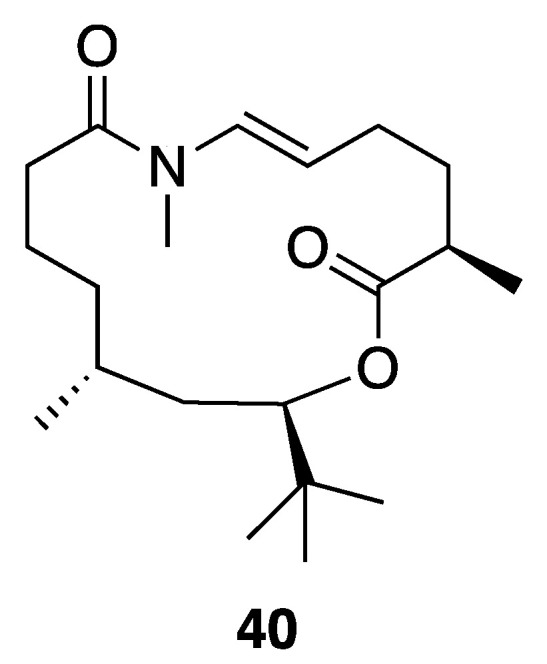
Palmyrolide A (**40**) from *Leptolyngbya cf.* sp.

**Figure 12 marinedrugs-18-00508-f012:**
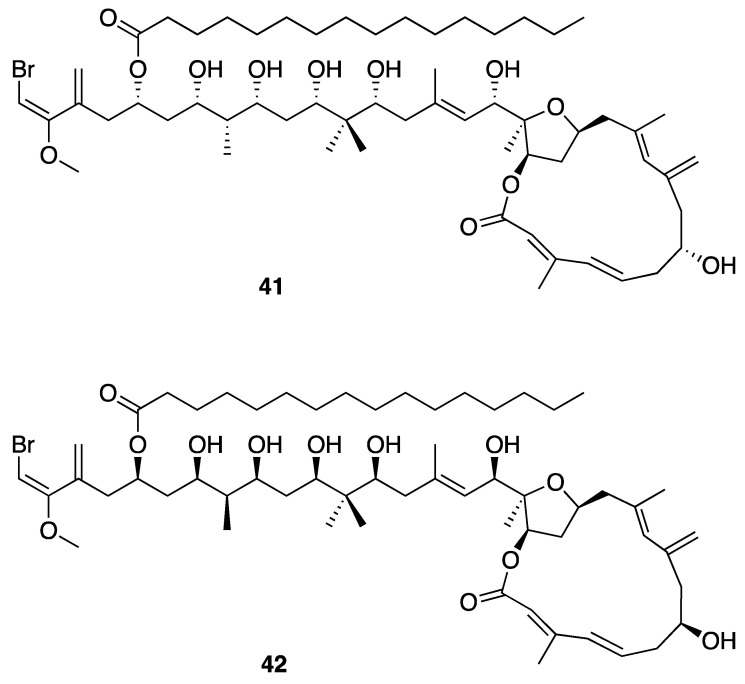
Phormidolide from *Leptolyngbya* sp. with the original proposed stereostructure (**41**) and revised stereostructure (**42**).

**Figure 13 marinedrugs-18-00508-f013:**
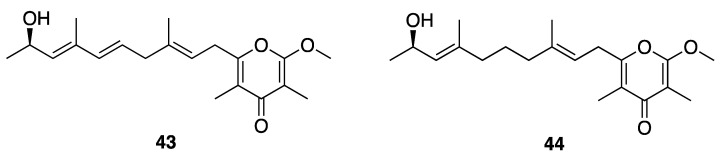
Kalkpyrones from *Leptolyngbya* sp.

**Figure 14 marinedrugs-18-00508-f014:**
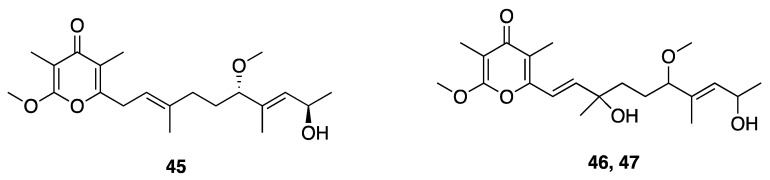
Yoshinones from *Leptolyngbya* sp.

**Figure 15 marinedrugs-18-00508-f015:**
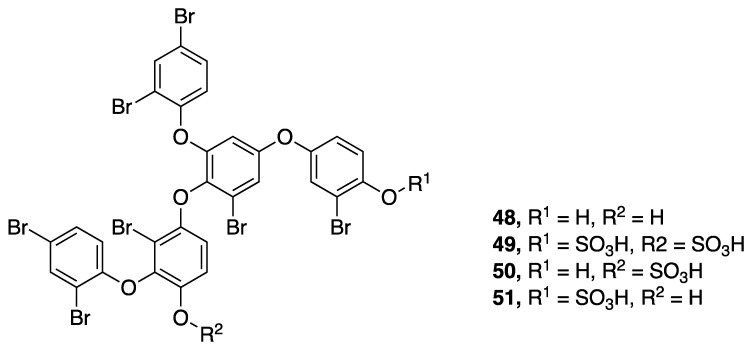
Crossbyanols from *Leptolyngbya crossbyana.*

**Figure 16 marinedrugs-18-00508-f016:**
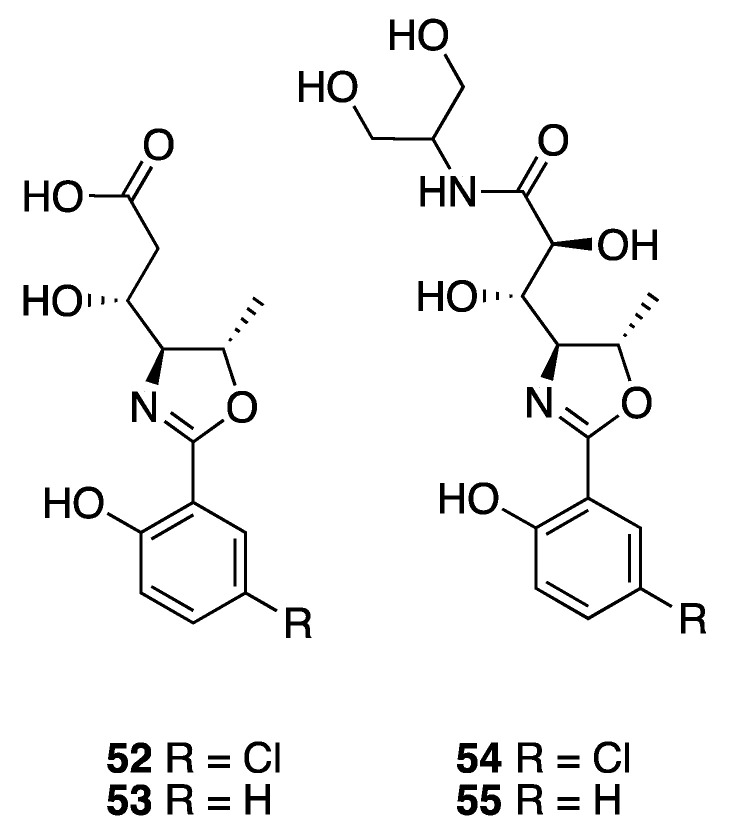
Leptazolines from *Leptolyngbya* sp.

**Figure 17 marinedrugs-18-00508-f017:**
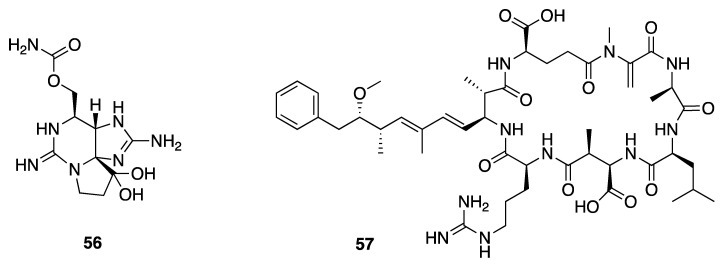
Two toxins, saxitoxin (**56**) and microcystin-LR (**57**), isolated from *Leptolyngbya* sp.

**Figure 18 marinedrugs-18-00508-f018:**
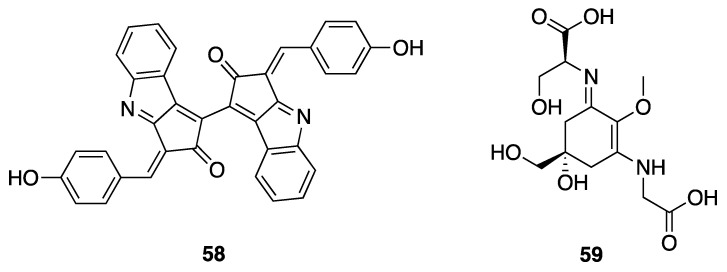
Scytonemin (**58**) and the MAA shinorine (**59**) from *Leptolyngbya* sp.

**Figure 19 marinedrugs-18-00508-f019:**
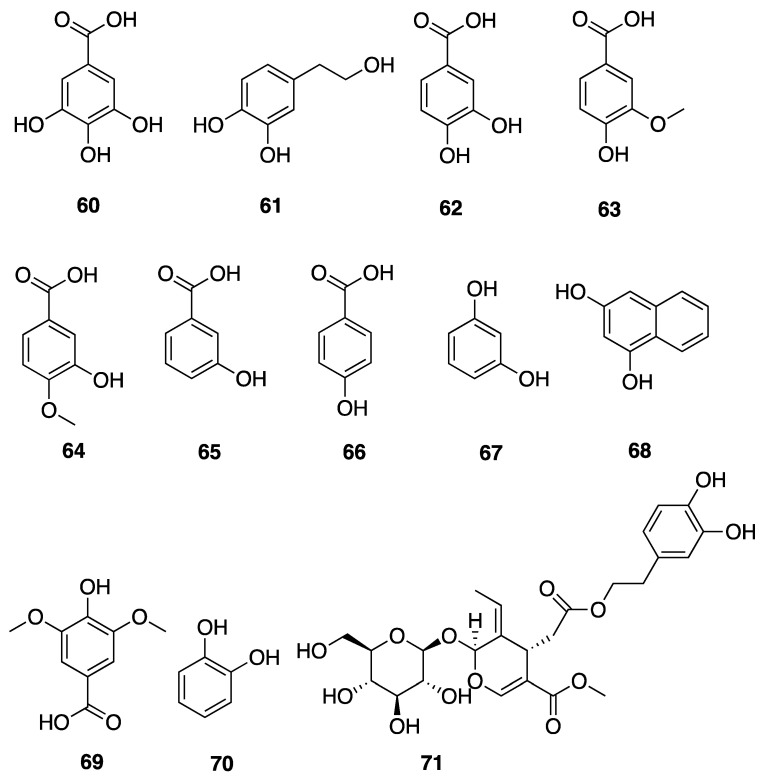
Hydroxybenzoic acids (HBAs) from *Leptolyngbya* sp.

**Figure 20 marinedrugs-18-00508-f020:**
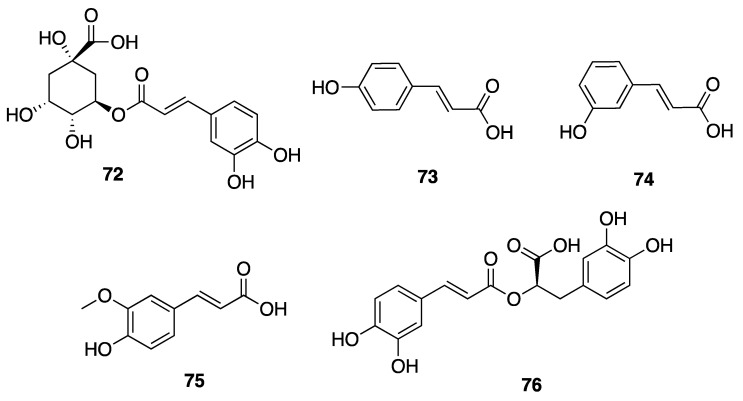
Hydroxycinnamic acids (HCAs) from *Leptolyngbya* sp.

**Figure 21 marinedrugs-18-00508-f021:**
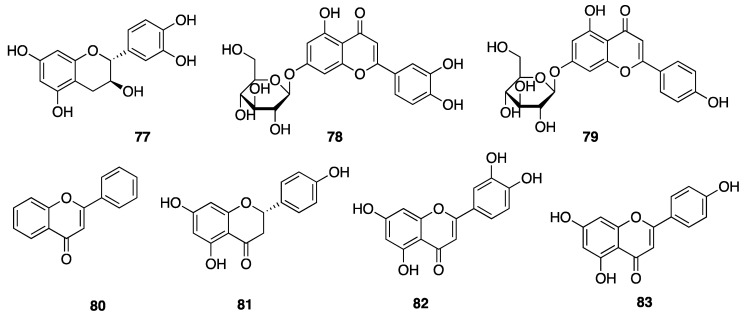
Flavonoids from *Leptolyngbya sp.*

**Figure 22 marinedrugs-18-00508-f022:**
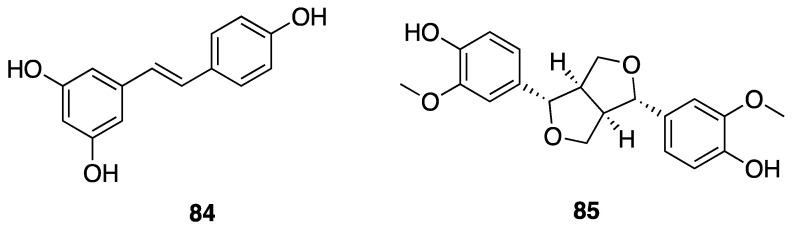
Stilbene and a lignan from *Leptolyngbya* sp.

**Figure 23 marinedrugs-18-00508-f023:**
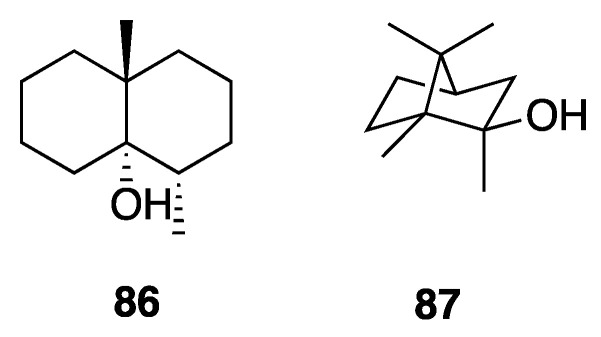
Geosmin and 2-methylisoborneol (MIB) from *Leptolyngbya bijugata.*

**Table 1 marinedrugs-18-00508-t001:** Secondary metabolites from *Leptolyngbya* and the reported bioactivities from each.

Name	Geographic Location	Culture	Total Synthesis	Bioactivity	Cell Line	Activity	Reference
Coibamide A (1)	N *^a^*	N *^a^*	Y *^b^*	Cytotoxicity	MDA-MB-231	IC_50_ 3.9 nM	[30]
				Cytotoxicity	A549	IC_50_ 3.6 nM	
				Cytotoxicity	MCF-7	IC_50_ 35.7 nM	
	N *^a^*	N *^a^*	Y *^b^*	Cytotoxicity	MDA-MB-231	GI_50_ 5.0 nM	[31]
				Cytotoxicity	A549	GI_50_ 5.4 nM	
				Cytotoxicity	PANC-1	GI_50_ 3.1 nM	
	Coiba National Park, Panama	N *^a^*	N *^a^*	Cytotoxicity	H460	LC_50_ < 23 nM	[32]
				Cytotoxicity	Mouse neuro2a	LC_50_ < 23 nM	
				Cytotoxicity	MDA-MB-231	GI_50_ 2.8 nM	
				Cytotoxicity	LOX IMVI	GI_50_ 7.4 nM	
				Cytotoxicity	HL-60(TB)	GI_50_ 7.4 nM	
				Cytotoxicity	SNB-75	GI_50_ 7.6 nM	
				Histological Selectivity	Breast, CNS, colon, and ovarian cancer cells	Good	
	Coiba National Park, Panama	N *^a^*	N *^a^*	Cytotoxicity	U87-MG	EC_50_ 28.8 nM	[33]
				Cytotoxicity	SF-295 glioblastoma cell	EC_50_ 96.2 nM	
				Cytotoxicity	MEFs	EC_50_ 96.2 nM	
Synthetic l-HVA, l-MeAla-Coibamide A (2)	N *^a^*	N *^a^*	Y *^b^*	Cytotoxicity	COLO205	IC_50_ 11.5 µM	[33]
				Cytotoxicity	H460	45% inhibition at 20 µM	
				Cytotoxicity	MDA-MB-231	IC_50_ 17.98 µM	[34]
				Cytotoxicity	MCF-7	IC_50_ 11.77 µM	
				Cytotoxicity	A549	IC_50_ 22.80 µM	
				Cytotoxicity	MDA-MB-231	GI_50_ > 16000 nM	[31]
				Cytotoxicity	A549	GI_50_ 22800 nM	
				Cytotoxicity	PANC-1	ND *^c^*	
	N *^a^*	N *^a^*	N *^a^*	Cytotoxicity	H292	IC_50_ 124 nM	[35]
				Cytotoxicity	MDA-MB-231	IC_50_ 66 nM	
				Cytotoxicity	PC-3	IC_50_ 80 nM	
				Cytotoxicity	SF-295	IC_50_ 219 nM	
Synthetic O-Desmethyl, l-HVA, l-MeAla-Coibamide A (3)	N *^a^*	N *^a^*	Y *^b^*	Cytotoxicity	COLO205	IC_50_ 13 µM	[33]
				Cytotoxicity	H460	36% inhibition at 20 µM	
Synthetic l-HVA, d-MeAla-Coibamide A (4)	N *^a^*	N *^a^*	Y *^b^*	Cytotoxicity	A549	IC_50_ 19.0 nM	[35]
				Cytotoxicity	HCT116	IC_50_ 44.6 nM	
				Cytotoxicity	MCF-7	IC_50_ 48.6 nM	
				Cytotoxicity	B16	IC_50_ 54.4 nM	
				Cytotoxicity	H292	IC_50_ 610 nM	
				Cytotoxicity	MDA-MB-231	IC_50_ 545 nM	
				Cytotoxicity	PC-3	IC_50_ 424 nM	
				Cytotoxicity	SF-295	IC_50_ 816 nM	
				Cytotoxicity	MDA-MB-231	GI_50_ 545 nM	[31]
				Cytotoxicity	A549	GI_50_ 19 nM	
				Cytotoxicity	PANC-1	ND *^c^*	
Synthetic Coibamide A-1c (5)	N *^a^*	N *^a^*	Y *^b^*	Cytotoxicity	MDA-MB-231	GI_50_ 7518 nM	[31]
				Cytotoxicity	A549	GI_50_ 20091 nM	
				Cytotoxicity	PANC-1	GI_50_ 12417 nM	
Synthetic Coibamide A-1d (6)	N *^a^*	N *^a^*	Y *^b^*	Cytotoxicity	MDA-MB-231	GI_50_ 10809 nM	[31]
				Cytotoxicity	A549	ND *^c^*	
				Cytotoxicity	PANC-1	ND *^c^*	
Synthetic Coibamide A-1e (7)	N *^a^*	N *^a^*	Y *^b^*	Cytotoxicity	MDA-MB-231	GI_50_ 2662 nM	[31]
				Cytotoxicity	A549	GI_50_ 1995 nM	
				Cytotoxicity	PANC-1	GI_50_ 1906 nM	
Synthetic MeAla3-MeAla6-Coibamide A-1f (8)	N *^a^*	N *^a^*	Y *^b^*	Cytotoxicity	MDA-MB-231	GI_50_ 5.1 nM	[31]
				Cytotoxicity	A549	GI_50_ 7.3 nM	
				Cytotoxicity	PANC-1	GI_50_ 7.0 nM	
Synthetic Coibamide A-1g (9)	N *^a^*	N *^a^*	Y *^b^*	Cytotoxicity	MDA-MB-231	GI_50_ 5.3 nM	[31]
				Cytotoxicity	A549	GI_50_ 12.4 nM	
				Cytotoxicity	PANC-1	GI_50_ 32.9 nM	
Synthetic Coibamide A-1h (10)	N *^a^*	N *^a^*	Y *^b^*	Cytotoxicity	MDA-MB-231	GI_50_ 61.6 nM	[31]
				Cytotoxicity	A549	GI_50_ 81.7 nM	
				Cytotoxicity	PANC-1	GI_50_ 124 nM	
Synthetic Coibamide A-1i (11)	N *^a^*	N *^a^*	Y *^b^*	Cytotoxicity	MDA-MB-231	GI_50_ 20.8 nM	[31]
				Cytotoxicity	A549	GI_50_ 194 nM	
				Cytotoxicity	PANC-1	GI_50_ 46.3 nM	
Synthetic Coibamide A-1j (12)	N *^a^*	N *^a^*	Y *^b^*	Cytotoxicity	MDA-MB-231	GI_50_ 2056 nM	[31]
				Cytotoxicity	A549	ND *^c^*	
				Cytotoxicity	PANC-1	GI_50_ 2178 nM	
Synthetic Coibamide A-1k (13)	N *^a^*	N *^a^*	Y *^b^*	Cytotoxicity	MDA-MB-231	GI_50_ 183 nM	[31]
				Cytotoxicity	A549	GI_50_ 222 nM	
				Cytotoxicity	PANC-1	GI_50_ 277 nM	
Synthetic Coibamide A-1l (14)	N *^a^*	N *^a^*	Y *^b^*	Cytotoxicity	MDA-MB-231	GI_50_ 450 nM	[31]
				Cytotoxicity	A549	GI_50_ 473 nM	
				Cytotoxicity	PANC-1	GI_50_ 601 nM	
Synthetic Coibamide A-1m (15)	N *^a^*	N *^a^*	Y *^b^*	Cytotoxicity	MDA-MB-231	GI_50_ 415 nM	[31]
				Cytotoxicity	A549	GI_50_ 511 nM	
				Cytotoxicity	PANC-1	GI_50_ 723 nM	
Synthetic Coibamide A-1n (16)	N *^a^*	N *^a^*	Y *^b^*	Cytotoxicity	MDA-MB-231	GI_50_ >16000nM	[31]
				Cytotoxicity	A549	ND *^c^*	
				Cytotoxicity	PANC-1	ND *^c^*	
Synthetic Coibamide A-1o (17)	N *^a^*	N *^a^*	Y *^b^*	Cytotoxicity	MDA-MB-231	GI_50_ 470 nM	[31]
				Cytotoxicity	A549	GI_50_ 733 nM	
				Cytotoxicity	PANC-1	GI_50_ 828 nM	
Synthetic Coibamide A-1p (18)	N *^a^*	N *^a^*	Y *^b^*	Cytotoxicity	MDA-MB-231	GI_50_ 236 nM	[31]
				Cytotoxicity	A549	GI_50_ 360 nM	
				Cytotoxicity	PANC-1	GI_50_ 204 nM	
Synthetic Coibamide A-1q (19)	N *^a^*	N *^a^*	Y *^b^*	Cytotoxicity	MDA-MB-231	GI_50_ 239 nM	[31]
				Cytotoxicity	A549	GI_50_ 443 nM	
				Cytotoxicity	PANC-1	GI_50_ 415 nM	
Synthetic Coibamide A-1r (20)	N *^a^*	N *^a^*	Y *^b^*	Cytotoxicity	MDA-MB-231	GI_50_ >16000 nM	[31]
				Cytotoxicity	A549	ND *^c^*	
				Cytotoxicity	PANC-1	ND *^c^*	
Dolastatin 12 (21)	The Red Sea	Y *^b^*	N *^a^*	Cytotoxicity	HeLa cells	IC_50_ > 1 μM	[36]
					KB (human nasopharyngeal carcinoma cell line)	MICs <0.05 µg/mL	[37]
					LoVo (a human colon adenocarcinoma cell line)	0.08 µg/mL	
					Murine P388 lymphocytic leukemia	ED_50_ 7.5 × 10^−2^ µg/mL	[38,39]

Ibu-Epidemethoxylyngbyastatin 3 (22)	The Red Sea	Y *^b^*	N *^a^*	Cytotoxicity	HeLa cells	IC_50_ > 10 μM	[36]
Grassypeptolide D (23)	The Red Sea	Y *^b^*	N *^a^*	Cytotoxicity	HeLa cells	IC_50_ 335 nM	[36]
				Cytotoxicity	Mouse neuro2a blastoma cells	IC_50_ 599 nM	
Grassypeptolide E (24)				Cytotoxicity	HeLa cells	IC_50_ 192 nM	[36]
				Cytotoxicity	Mouse neuro2a blastoma cells	IC_50_ 407 nM	
Loggerpeptin A (25)	Florida, USA	N *^a^*	N *^a^*	Antiproteolytic Activity	Bovine pancreatic chymotrypsin	IC_50_ 0.24 μM	[40]
					Porcine pancreatic elastase	IC_50_ 0.24 μM	
					Human neutrophil elastase	IC_50_ 0.29 μM	
Loggerpeptin B (26)	Florida, USA	N *^a^*	N *^a^*	Antiproteolytic Activity	Bovine pancreatic chymotrypsin	IC_50_ 0.22 μM	[40]
					Porcine pancreatic elastase	IC_50_ 0.28 μM	
					Human neutrophil elastase	IC_50_ 0.89 μM	
Loggerpeptin C (27)	Florida, USA	N *^a^*	N *^a^*	Antiproteolytic Activity	Bovine pancreatic chymotrypsin	IC_50_ 0.35 μM	[40]
					Porcine pancreatic elastase	IC_50_ 0.54 μM	
					Human neutrophil elastase	IC_50_ 0.62 μM	
Molassamide (28)	Florida, USA	N *^a^*	N *^a^*	Antiproteolytic Activity	Bovine pancreatic chymotrypsin	IC_50_ 0.24 μM	[40]
					Porcine pancreatic elastase	IC_50_ 0.05 μM	
					Human neutrophil elastase	IC_50_ 0.11 μM	
2-Hydroxyethyl-11-Hydroxyhexadec-9-Enoate (29)	Gulf of Thailand	Y *^b^*	N *^a^*	Antibacterial Activities	*Vibrio harveyi*	MIC 250–1000 µg/mL	[41]
				Antibacterial Activities	*Vibrio parahaemolyticus*	MIC 350–1000 µg/mL	
Honaucin A (30)	Hawaii, USA	N *^a^*	N *^a^*	Anti-Inflammatory Activity	LPS-stimulated RAW264.7 murine macrophages	IC_50_ 4.0 µM	[42]
				Antioxidant Activity	Radical Oxygen Scavenging	No activity at 146 µM	
				QS-Inhibitory activities	*V. harveyi* BB120	IC_50_ 5.6 µM	
				QS-Inhibitory activities	*E. coli* JB525	IC_50_ 38.5 µM	
				Cytotoxicity	RAW264.7 cells	No activity at 1 µg/mL	[43]
				Cellular TRAP Activity	RANKL-induced osteoclastogenesis in RAW264.7 cells	IC_50_ 0.63 μg/mL	
Honaucin B (31)	Hawaii, USA	N *^a^*	N *^a^*	Anti-Inflammatory Activity	LPS-stimulated RAW264.7 murine macrophages	IC_50_ 4.5 µM	[42]
				QS-Inhibitory activities	*V. harveyi* BB120	IC_50_ 17.6 µM	
				QS-Inhibitory activities	*E. coli* JB525	IC_50_ > 500 µM	
Honaucin C (32)	Hawaii, USA	N *^a^*	N *^a^*	Anti-Inflammatory Activity	LPS-stimulated RAW264.7 murine macrophages	IC_50_ 7.8 µM	[42]
				QS-Inhibitory activities	*V. harveyi* BB120	IC_50_ 14.6 µM	
				QS-Inhibitory activities	*E. coli* JB525	IC_50_ > 500 µM	
Synthetic Br-Honaucin A (33)	N *^a^*	N *^a^*	Y *^b^*	Cytotoxicity	RAW264.7 cells	No activity at 1 µg/mL	[43]
				Cellular TRAP Activity	RANKL-induced osteoclastogenesis in RAW264.7 cells	IC_50_ 0.54 μg/mL	
Synthetic Hex-Honaucin A (34)	N *^a^*	N *^a^*	Y *^b^*	Cytotoxicity	RAW264.7 cells	71.6% cell viability at 1 µg/mL	[43]
				Cellular TRAP Activity	RANKL-induced osteoclastogenesis in RAW264.7 cells	IC50 0.68 μg/mL	
Synthetic I-Honaucin A (35)	N *^a^*	N *^a^*	Y *^b^*	Cytotoxicity	RAW264.7 cells	No activity at 1 µg/mL	[43]
				Cellular TRAP Activity	RANKL-induced osteoclastogenesis in RAW264.7 cells	IC_50_ 0.61 μg/mL	
Leptolyngbyolide A (36)	Okinawa, Japan	N *^a^*	Y *^b^*	Cytotoxicity	HeLa S3 cell	IC_50_ 0.099 µM	[44]
				Actin-Depolymerizing activity	F-actin	EC_50_ 12.6 µM	
Leptolyngbyolide B (37)	Okinawa, Japan	N *^a^*	Y *^b^*	Cytotoxicity	HeLa S3 cell	IC_50_ 0.16 µM	[44]
				Actin-Depolymerizing activity	F-actin	EC_50_ 11.6 µM	
Leptolyngbyolide C (38)	Okinawa, Japan	N *^a^*	Y *^b^*	Cytotoxicity	HeLa S3 cell	IC_50_ 0.64 µM	[44]
				Actin-Depolymerizing activity	F-actin	EC_50_ 26.9 µM	
Leptolyngbyolide D (39)	Okinawa, Japan	N *^a^*	Y *^b^*	Cytotoxicity	HeLa S3 cell	IC_50_ 0.15 µM	[44]
				Actin-Depolymerizing activity	F-actin	EC_50_ 21.5 µM	
Palmyrolide A (40)	Palmyra Atoll	N *^a^*	Y *^b^*	Ca^2+^ Influx (Inhibition)	Murine cerebrocortical neurons	IC_50_ 3.70 µM (2.29–5.98 µM, 95% CI)	[45]
				Na^+^ Channel Blocking Activity	Mouse neuroblastoma (neuro2a)	IC_50_ 5.2 µM	
				Cytotoxicity	H460	No activity at 20 µM	
Phormidolide (42)	The Red Sea	Y *^b^*	N *^a^*	Brine Shrimp Toxicity		LC_50_ 1.5 µM	[46]
Kalkipyrone A (43)	America Samoa	N *^a^*	N *^a^*	Cytotoxicity	H460 cells	EC_50_ 0.9 µM	[47]
				Cytotoxicity	*Saccharomyces cerevisiae* ABC16-monster	IC_50_ 14.6 µM	
				Brine Shrimp Toxicity	Brine shrimp (*Artemia salina*)	LD_50_ 1 µg/mL	[48]
				Ichthyotoxicity	Goldfish *Carassius auratus*	LD_50_ 2 µg/mL	
				Cytotoxicity	NCI’s 60 human tumor cell line	Modestly inhibitory to several renal and melanoma cell lines	
Kalkipyrone B (44)	America Samoa	N *^a^*	N *^a^*	Cytotoxicity	H460 cells	EC_50_ 9.0 µM	[47]
				Cytotoxicity	*Saccharomyces cerevisiae* ABC16-monster	IC_50_ 13.4 µM	
Yoshinone A (45)	Ishigaki island, Japan	N *^a^*	Y *^b^*	Adipogenic Differentiation	3T3-L1 cells	EC_50_ 420 nM	[49]
				Cytotoxicity	3T3-L1 cells	IC_50_ > 50 µM	
				Cytotoxicity	HeLa	IC_50_ > 50 µM	
				Cytotoxicity	*Saccharomyces cerevisiae* ABC16-monster	IC_50_ 63.8 µM	[47]
				Cytotoxicity	H460 cells	EC_50_ > 10 µM	
Yoshinone B1 (46)	Ishigaki island, Japan	N *^a^*	N *^a^*	Adipogenic Differentiation	3T3-L1 cells	<50% inhibition at 5 µM	[49]
Yoshinone B2 (47)	Ishigaki island, Japan	N *^a^*	N *^a^*	Adipogenic Differentiation	3T3-L1 cells	<50% inhibition at 5 µM	[49]
Crossbyanol A (48)	Hawaii, USA	N *^a^*	N *^a^*	Cytotoxicity	H460 human lung cancer cells	IC_50_ 30 µg/ mL	[50]
				Na^+^ Influx (Activation and Inhibition)	Mouse neuroblastoma (neuro2a)	IC_50_ 20 µg/mL(Activation)	
				Antibacterial Activity	Methicillin-resistant *Staphylococcus aureus* (MRSA)	No activity at 125 µg/mL	
				Brine Shrimp Toxicity	Brine shrimp (*Artemia salina*)	No activity at 25 µg/mL	
Crossbyanol B (49)	Hawaii, USA	N *^a^*	N *^a^*	Cytotoxicity	H460 human lung cancer cells	No activity at 20 µg/mL	[50]
				Na^+^ Influx (Activation and Inhibition)	Mouse neuroblastoma (neuro2a)	No activity at 20 µg/mL	
				Antibacterial activity	Methicillin-resistant *Staphylococcus aureus* (MRSA)	MIC 2.0–3.9 µg/mL	
				Brine Shrimp Toxicity	Brine shrimp (*Artemia salina*)	IC_50_ 2.8 µg/mL	
Crossbyanol C (50)	Hawaii, USA	N *^a^*	N *^a^*	Cytotoxicity	H460 human lung cancer cells	No activity at 20 µg/mL	[50]
				Na^+^ Influx (Activation and Inhibition)	Mouse neuroblastoma (neuro2a)	No activity at 20 µg/mL	
Crossbyanol D (51)	Hawaii, USA	N *^a^*	N *^a^*	Cytotoxicity	H460 human lung cancer cells	No activity at 20 µg/mL	[50]
				Na^+^ Influx (Activation and Inhibition)	Mouse neuroblastoma (neuro2a)	No activity at 20 µg/mL	
Leptazoline A (52)	Honolulu	Y *^b^*	N *^a^*	Cytotoxicity	PANC-1	No significant activity	[51]
Leptazoline B (53)	Honolulu	Y *^b^*	N *^a^*	Cytotoxicity	PANC-1	GI_50_ 10 µM	

*^a^* Not found in literature. *^b^* Found in literature. *^c^* Not determined.

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
