# Peer review of "The Chemistry, Biochemistry and Pharmacology of Marine Natural Products from Leptolyngbya, a Chemically Endowed Genus of Cyanobacteria"

_marinedrugs, 2020, doi:10.3390/md18100508_

Round 1

Reviewer 1 Report

The work, in a comprehensive way, presents the existing knowledge about bioactive metabolites produced by cyanobacteria of the genus Leptolyngbya. Although these cyanobacteria commonly occur in different water bodies worldwide, and their potential as a source of lead compounds for drug development has been well documented, the authors consider this genus to be underexplored. At least in part, it might be due to difficulties in growing Leptolyngbya for biomass in culture.

My specific comments are as follows:

As in case of cyanobacteria, significant inter- and intra-species differences in the produced metabolites exist, I would expect that (where available) the exact names/numbers of the strains (sources of the metabolite) will be given.

The work is not free from typographical error.

  • In case of cyanobacterial genera (e.g. Leptolyngbya), “sp.” should not be in italic, while in references, the genus/species names should be written in italic.
  • Line 55: “are” should be deleted.
  • Line 107-109: The sentence should be reworded, I think. It’s unclear.
  • Line 129 and 130: MAP and Ibu have already been explained in lines 110 and 113.
  • Line 177-178: Check the sentence.
  • Line 223: Probably “its”, instead of “it is”
  • Line 255: Probably “was found to activate” (without “have”)
  • Line 280: Why did the authors mention just one microcystin analogue (MC-LR). If no specific analogues of saxitoxins are given, the same should be in case of microcystins.

It is a well written and interesting work and I recommend it for publishing in Marine Drugs – after minor revision.

Reviewer 2 Report

This is very useful and interesting review paper. I only have one point for improving this review paper.

Authors mentioned Biochemistry in their title, but there are not much explanations in each section. Although Biochemistry seems to be obscure, but, synthetic pathway with corresponding enzymes in biosynthesis pathway should be explained in some cases of metabolites.

Also, authors should explain some cases regarding drug-target molecular interaction in terms of structure-activity relationship study at the amino acid levels in binding sites of target protein. Explaining mode of action against target enzyme activity with modeling results should be included.

Round 2

Reviewer 2 Report

I accept current revision.